# Building Orientation in Green Facade Performance and Its Positive Effects on Urban Landscape Case Study: An Urban Block in Barcelona

**Faezeh Bagheri Moghaddam [1],\*, Josep Maria Fort Mir [1], Alia Besné Yanguas [2], Isidro Navarro Delgado [1] and Ernest Redondo Dominguez [1]**

[1] Escola Tècnica Superior d'Arquitectura de Barcelona, Universitat Politècnica de Catalunya, 08028 Barcelona, Spain; Josep.Maria.Fort@upc.edu (J.M.F.M.); Isidro.Navarro@upc.edu (I.N.D.); Ernesto.Redondo@upc.edu (E.R.D.)

[2] Escola Tècnica Superior d'Arquitectura La Salle, Universitat Ramon Llull, 08022 Barcelona, Spain; alia.besne@salle.url.edu

\* Correspondence: FFaezeh.Bagheri.Moghaddam@upc.edu

**Abstract:** This paper addresses the effect of building orientation efficiency of the green facade in energy consumption, for which the case study is an urban block in Passeig de Gracia, L'Eixample, Barcelona. Nowadays, many countries are faced with the trouble of the deficiency of energy resources and the incapability of saving them. Most of this energy is consumed in the cooling, heating, and artificial ventilation of buildings. For this reason, the development of an integrated strategy like a green facade is essential to transform buildings into structures that consume less energy and to improve the occupants' comfort conditions. From the perspective of the urban landscape, the green facade can influence the quality of life in cities due to its positive effects such as the purification of air, the absorption of carbon dioxide, and the mitigation of dust, as well as the aesthetic and psychological aspects. Such criteria are based on the adoption of suitable orientation for the green facade, which is the second layer of the facade in an office building with a curtain wall as the main facade. Since the most important factor in the implementation of a green facade is the building's orientation, the optimum orientation could be the key factor in regards to the reduction of energy consumption and cost and the improvement of overall energy efficiency. We used software that helped simulate the total energy consumption, the cost, and the energy use intensity annually and monthly. Consequently, after testing was carried out, it was proven that a green facade as a second layer with a southeast and/or a southwest orientation results in the maximum energy saving in a coastal city with a Mediterranean climate like Barcelona.

**Keywords:** vertical garden; green facade; building orientation; energy consumption; sustainability; urban landscape; simulation software

## 1. Introduction

In recent decades, countries have faced plenty of issues related to energy supplies and the effects of global warming and urban heat islands (UHIs) on energy consumption [1]. For this reason, architects and urban planners have proposed a newer design approach, namely the sustainable building design, to reduce the heat island effect and energy demand and minimize environmental effects [2]. The green facade is an element of sustainable building design which is gradually gaining popularity, and it is being applied extensively on a large scale [3,4]. Moreover, using plants in the facade (green facade) is a bioclimatic strategy that would be effective in reducing energy consumption in buildings, in addition to other psychological, aesthetic, and economic benefits [5].

Many studies have revealed the positive effects of the adoption of the green facade in buildings and those buildings' orientation on energy consumption efficiency [6].

A building with the right orientation can double the efficiency of the green facade as a second layer in the facade [1,2]. Utilizing the appropriate building orientation when applying a vertical garden could save a lot of money as it would no longer require heating and cooling expenditure costs; in fact, the building itself would provide a comfortable environment for occupants through energy reduction and cost reduction [3–5]. By using a green facade, occupants can reduce heating and cooling consumption. An extra benefit is that there is nothing that can fail or break down with a building that has the appropriate orientation for the application of a green layer in the building's facade; as a result, this strategy is called "passive solar" [6] due to the almost zero maintenance costs that could be incurred during the lifetime of the green facade. It is important to note that the choice of plants is to be taken into account as they must be suitable for the specific orientation of the building for such a facade to be successful. For example, a building orienting south must opt for sun-resistant plants [7,8].

Building orientation has been one of the primary considerations within construction for thousands of years in many cultures. One of the original references for building orientation and passive solar principals was by Socrates about 2300 years ago [6]. "Now in houses with a south aspect, the sun's rays penetrate the porticos in winter, but in the summer the path of the sun is right over our heads and above the roof so that there is shade. If then, this is the best arrangement, we should build the south side loftier to get the winter sun and the north side lower to keep out the winter winds."

Pérez et al. [4] summed up the green facades mechanisms when used as a passive system for energy savings: the shadowing effect of the vegetation shields the building's surface from solar radiation, and vegetation also provides thermal insulation, as when the plants' evapotranspiration occurs, the evaporative cooling in the substrate and the effect of the wind on the building change.

Nowadays, many countries have adopted different construction methods to obtain benefits from solar radiation and building orientations, like double skin and green facade as a second skin [2], especially in glass facades. In fact, it was discovered that building behavior in response to solar radiation could be changed in different climates by implementing passive solutions [9]. One way to reinforce passive solutions in buildings is to implement a green facade as a second layer in buildings, especially in Mediterranean climates as they would benefit the most from an environment without artificial devices [8].

In fact, one factor that causes the growth of a building's energy consumption is high temperatures, because they result in intolerable cooling demand [10–16]. It is estimated that midlatitude and temperate climates will face a significant increase in annual energy consumption because of climate change and urban heat island (UHI) scenarios as cooling will be required in autumn and spring as well [17,18].

The concept of building energy efficiency is related to the energy supply required which achieves suitable environmental conditions that could allow the reduction of energy consumption [19]. One of the best methods to reduce the cost of energy in buildings is a suitable heating and cooling design [20]. Variables of design and construction parameters should be optimized to design energy-efficient buildings [21]. Parameters that affect building energy requirements have been summarized by Ekici and Aksoy [22] (Table 1).

**Table 1.** Parameters that determine building energy requirements [22,23].

| Physical–Environmental Parameters | Design Parameters |
|---|---|
| | Shape factor |
| Daily outside temperature (°C) | Transparent surface |
| Solar radiation (W/m$^2$) | Orientation |
| Wind direction and speed (m/s) | Thermal–physical properties of building materials |
| | Distance between buildings |

In terms of urbanism, the green facade is one of the strategic implementations of urban green infrastructure (UGI) that can help urban landscape areas to achieve temperature reductions, causing the reduction of energy use within urban buildings, and it also has the added benefits of pollution reduction and the improvement of habitat biodiversity [24]. In high-density cities, the green facade could contribute to stress recovery and well-being, so the residents could benefit physiologically and psychologically from this UGI strategy [25].

The aim of this research was to investigate the impact of building orientation for a green facade on energy consumption. This paper presents a detailed description of the steps to take in order to benefit from the green facade as a second layer and its optimum orientation in Passeig de Gracia, L'Eixample area in Barcelona, Spain, by employing Autodesk Green Building Studio as a simulation software to prove the ability of the Green Building Studio to design high-performance buildings at a fraction of the time and cost of conventional methods [26,27].

## 2. Methodology

The methodology is based on the study of reducing energy consumption by applying green facades in different orientations, which causes an effect on the building's behavior. In addition, we discuss different strategies and architectural solutions to understand the reduction of energy consumption in buildings that have a green facade. Through the analysis of the previous research which explored the performance of a green facade by using a building simulation, we concluded that the structure and cavity depth in the application of the green facade are of great importance in regards to energy consumption reduction. For the first part, we selected an appropriate orientation (southeast), and we simulated a structure with different cavity depths. As a second simulation, we tested eight buildings with different orientations and specific cavity depths to understand the influence of different orientations on green facade performance.

To compare and observe the impact of this study, a single-skin run was added for each simulation. This is the advantage of using Green Building Studio, as it can recreate many simulations in one project, making it easy to compare the results in this case. The data created by the initial base run (no changes made in Green Building Studio and applied project default) were used for tests 1 to 6 with different cavity sizes and also in tests 1 to 8 which simulated different orientations.

### 2.1. Case Study and Scenario Descriptions

The scenario considering the green facade is generic; the application has a more complex building configuration. It was carried out in a green building design in Passeig de Gracia (street), L'Eixample area, in Barcelona (this area was designed by Ildefonso Cerdá in 1856) [28] (Figure 1). According to urban planning in Barcelona, each urban block has a 45° angle. The urban texture is continuous, dense, and compact; the average height of buildings ranges from 15 to 30 m. Given the different ages of planning, the size of the urban block varies within the city [15].

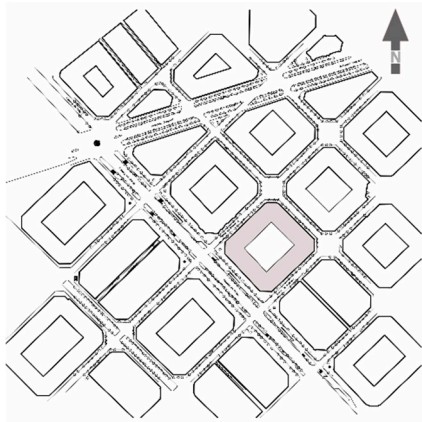
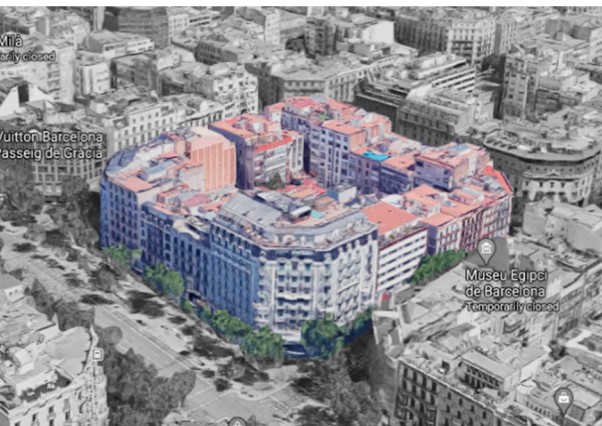

**Figure 1.** The case study is an urban block which is located in Passeig de Gracia, L'Eixample, in Barcelona. © By Author.

The case study is conceptual with cubic shape and a square plan in dimensions 10 × 10 m, 10 m high (Figures 2 and 3).

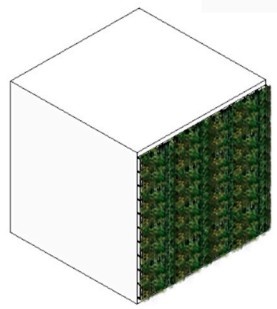

**Figure 2.** The morphology of the case study (10 × 10 × 10 m). © By Author.

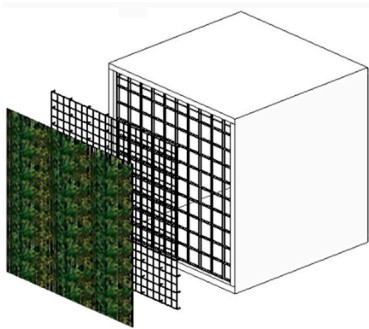

**Figure 3.** The layers of the facade. © By Author.

The main facade in this case study is a nonstructural curtain wall used only to separate the indoors from the outdoor weather. The curtain wall frame attaches to the building structure and does not carry the floor or roof loads. Regarding the methodology, the facade was considered in two simulations, and the first simulation included six tests, where test 1 was just a single skin (curtain wall) and tests 2 to 6 were green skins within a 10 to 50 cm cavity depth (see Table 2). This green facade is part of the facade that supports the green wall (horizontal aluminum slats) as a second layer that is applied to the facade. According to the classification of green walls that considers the horizontal aluminum slats as the continuous guides of an indirect green facade, this is a kind of green facade structure [29,30] (see Figure 4).

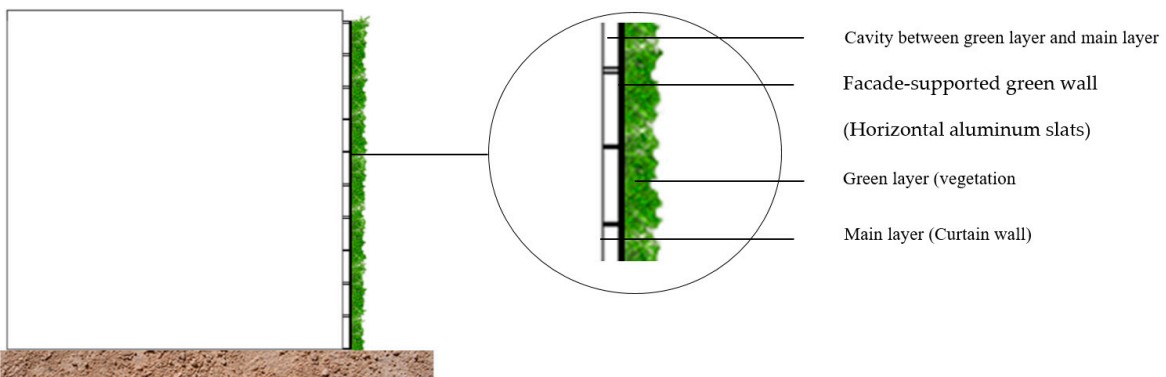

**Figure 4.** The implementation of the green layer on the facade with a cavity depth. © By author.

**Table 2.** The number of simulations with the same orientation but different sizes of the cavity depth. © By author.

| Test | Facade Type | | Cavity | Building Orientation | Facade Structure |
|------|-------------|--------------|--------|----------------------|------------------|
|      | **Single Layer** | **Second Layer** | | | |
| 1 | Single-skin facade | - | 0 | Southeast | Curtain wall (main facade) |
| 2 | - | Green-skin facade | 10 cm | Southeast | Facade-supported green wall (horizontal aluminum slats) |
| 3 | - | Green-skin facade | 20 cm | Southeast | Facade-supported green wall (horizontal aluminum slats) |
| 4 | - | Green-skin facade | 30 cm | Southeast | Facade-supported green wall (horizontal aluminum slats) |
| 5 | - | Green-skin facade | 40 cm | Southeast | Facade-supported green wall (horizontal aluminum slats) |
| 6 | - | Green-skin facade | 50 cm | Southeast | Facade-supported green wall (horizontal aluminum slats) |

## 3. Results

The results are divided into two parts. The first section shows the energy consumed within the different sizes of the cavity in the green layer of the facade. The second section presents simulation results for energy consumed in different orientations through eight tests.

*3.1. Analysis of the Energy Consumed with Different Cavity Depth Sizes in the Green Layer in Facade*

By using the simulation program, the energy consumption was studied and analyzed for each of the five different cavities in the green facade and compared with the single-skin facade (curtain wall) as the main facade with a southeast orientation in L'Eixample area of Barcelona throughout one year, as shown in Table 3.

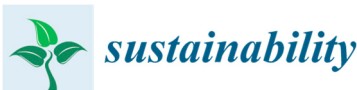 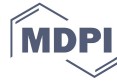

**Table 3.** Comparison of energy consumption with different cavity depths in southeast orientation. © By author.

| Name | Floor Area (m²) | Energy Use Intensity (MJ/m²/year) | Electric Cost (/kWh) | Fuel Cost (/MJ) | Total Annual Cost | | | Total Annual Energy | | |
|---|---|---|---|---|---|---|---|---|---|---|
| | | | | | Electric | Fuel | Energy | Electric (kWh) | Fuel (MJ) | Carbon Emissions (Mg) |
| Green Skin 50cm Cavity | 91 | 1,063.4 | € 0.13 | € 0.01 | € 1,675 | € 568 | € 2,243 | 13,397 | 48,897 | -- |
| Green Skin 40cm Cavity | 91 | 1,063.6 | € 0.13 | € 0.01 | € 1,644 | € 579 | € 2,223 | 13,150 | 49,811 | -- |
| Green Skin 30cm Cavity | 91 | 1,064.7 | € 0.13 | € 0.01 | € 1,668 | € 572 | € 2,240 | 13,342 | 49,216 | -- |
| Green Skin 20cm Cavity | 91 | 1,045.6 | € 0.13 | € 0.01 | € 1,613 | € 570 | € 2,183 | 12,901 | 49,063 | -- |
| Green Skin 10cm Cavity | 91 | 1,053.1 | € 0.13 | € 0.01 | € 1,602 | € 582 | € 2,184 | 12,817 | 50,045 | -- |
| Single Skin | 91 | 1,081.3 | € 0.13 | € 0.01 | € 2,247 | € 396 | € 2,643 | 17,974 | 34,062 | -- |

Table 3 shows that the optimum cavity size for this orientation (southeast) was 20 cm because it reduced the total energy cost (annual), the energy use intensity (EUI), and the total annual electricity use. However, fuel consumption was increased because of the decreased effect of sunlight due to the covering of the facade with the vegetation. Nevertheless, it should be noted that nowadays most heating and cooling devices, as well as lighting and air conditioning systems, use electrical energy. As a result, reducing electricity consumption is the most effective way to reduce energy consumption.

### 3.2. Analysis of the Energy Consumed in Different Orientations

After analyzing the first simulation (analysis of the energy consumed with different cavity depth sizes in the green layer of the facade), a 20 cm cavity depth size was chosen for the second simulation. In this section, we simulated the green facade building in different orientations with a 20 cm cavity depth (Table 4). As can be seen in Table 4, the building consumed more electricity for cooling in July, August, and September than in other months, and, by applying a green layer on the facade, the usage of electricity was reduced in all cases but the amount of reduction was different depending on the building's orientation. The most important data extracted from the simulation software were cooling and heating consumption; other energy consumption indicators like pumping or boiling water were not relevant for this research.

**Table 4.** Energy consumption comparison between a single-skin facade (curtain wall) and a green-skin facade in different orientations. © By author.

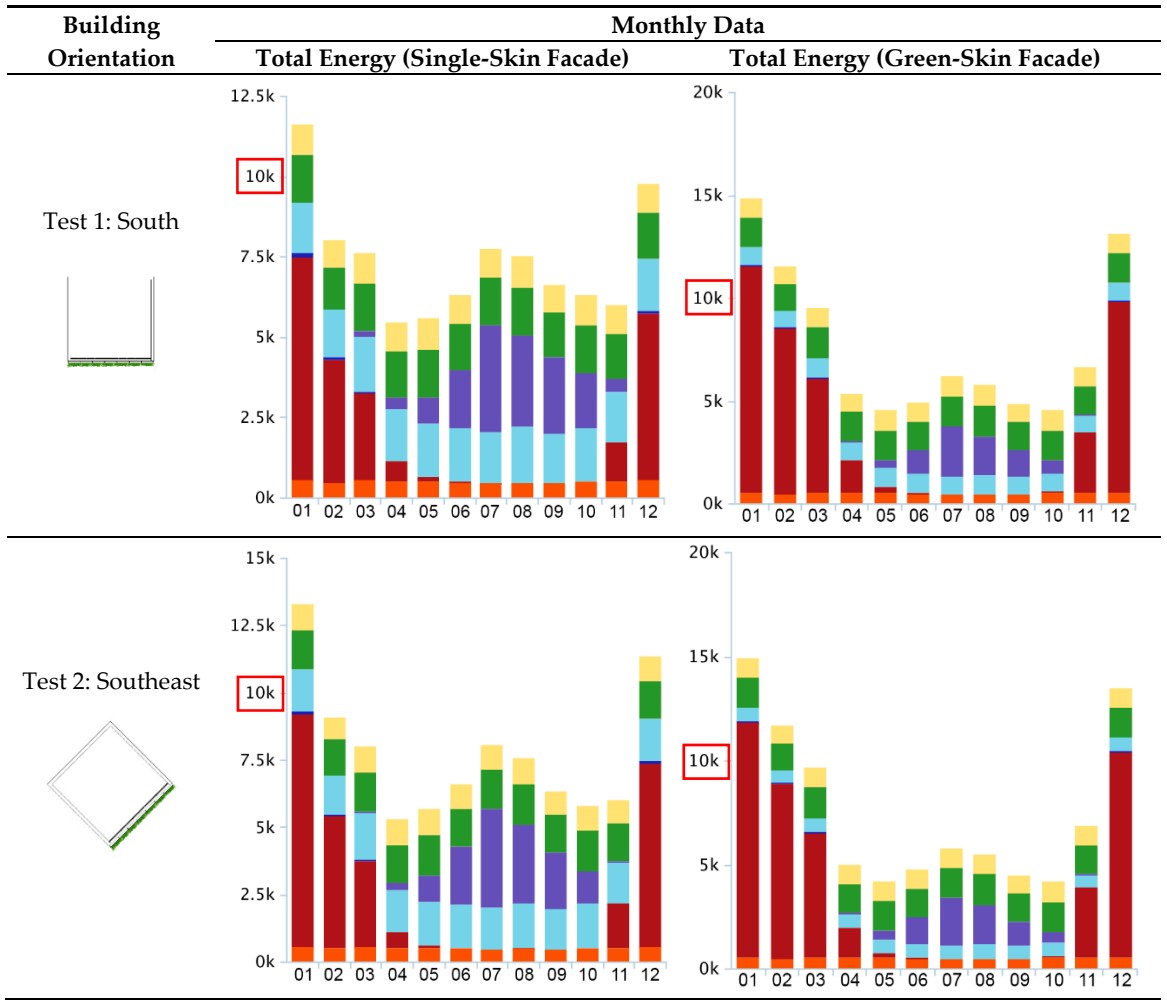

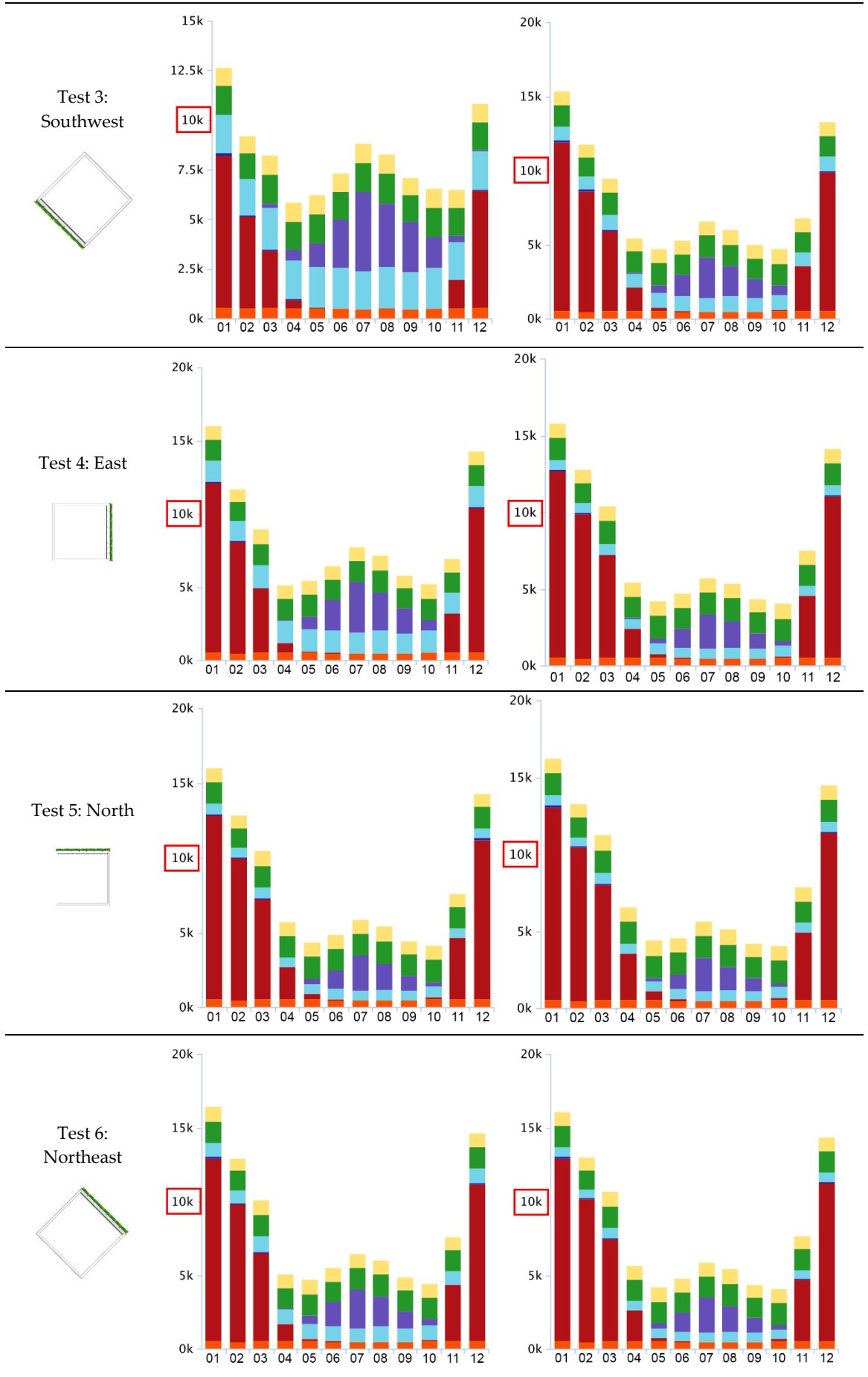

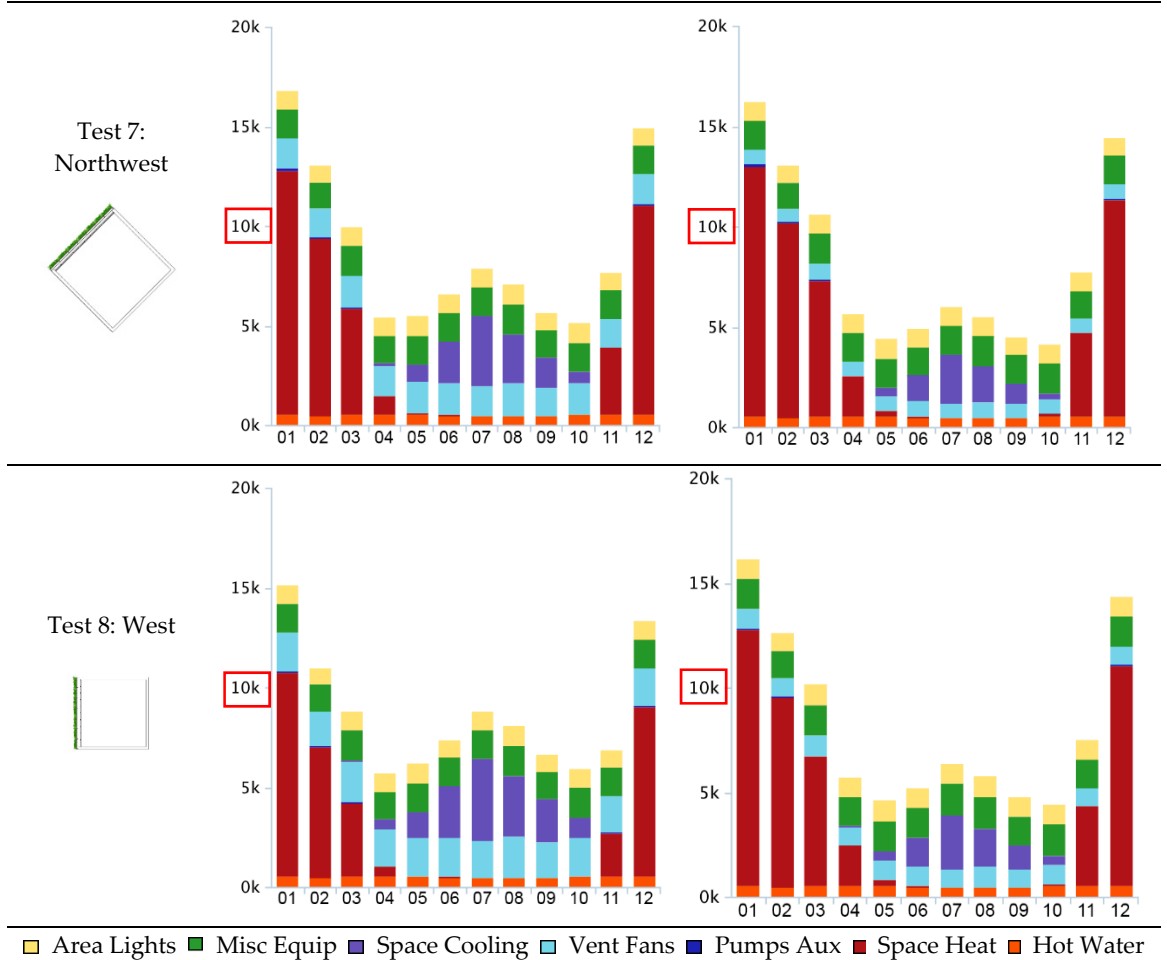

According to Table 5, which expresses the importance of building orientation in the performance of the green facade by comparing eight different orientations for the green facade, the green facade's performance varied from one orientation to another regarding the reduction of energy consumption. The southeastern green facade had the best performance in the reduction of energy use, especially in electrical energy, whereas the highest use of energy among orientations was found for the western green facade.

**Table 5.** Energy consumption and cost varying between eight different orientations. © By author.

| | | | Energy Consumption at Eight Orientations | | | | | | | |
|---|---|---|---|---|---|---|---|---|---|---|
| | | | South | South east | South west | East | North | North east | North west | West |
| Total Annual Energy Cost (€) | | Single Facade | €2600 | €2633 | €2863 | €2641 | €2242 | €2403 | €2711 | €2888 |
| | | Green Facade | €2273 | €2183 | €2351 | €2214 | €2232 | €2239 | €2282 | €2352 |
| Total Annual Energy | Electric (KWh) | Single Facade | 18,177 | 17,918 | 20,045 | 16,914 | 12,820 | 14,346 | 17,023 | 19,393 |
| | | Green Facade | 13,782 | 12,880 | 14,364 | 12,692 | 12,441 | 12,766 | 13,111 | 13,891 |
| | Fuel (MJ) | Single Facade | 28,209 | 33,832 | 30,734 | 45,314 | 12,820 | 52,445 | 50,137 | 39,883 |
| | | Green Facade | 47,365 | 49,287 | 47,824 | 53,952 | 58,220 | 55,378 | 55,335 | 52,965 |
| Energy Use Intensity (MJ/m²/year) | | Single Facade | 1025.3 | 1076.6 | 1126.5 | 1162.7 | 1107.7 | 1139.6 | 1219.8 | 1201.0 |
| | | Green Facade | 1061.7 | 1047.3 | 1089.7 | 1090.9 | 1127.8 | 1109.4 | 1122.6 | 1127.4 |

By considering the simulation of a green building in different orientations performed in this paper, it can be determined that the green facade's performance in regards to energy reduction results in different outcomes when angled at different orientations (Table 6). The northern and western green facades had a shortage of sun radiance, reducing the electrical use slightly and thus causing the use of energy for heating during winter and part of autumn and spring to not be sustainable. Such orientations obtain minimal performance of the green facade. In contrast, the total annual electrical consumption and cost in green facade buildings facing a southwest and/or a southeast orientation dropped significantly; this was thanks to solar energy, which has proven very effective for the green facade, that was captured by such orientations. These orientations use the maximum ability of the green facade for energy consumption, which can also be called passive energy. The green facade also provides shade, which reduces the use of cooling devices during hot weather; the second layer also protects the building during the cold weather and wind, consequently causing a change of building behavior.

**Table 6.** Annual electric and fuel end-use comparison between two types of facade (green and single skin) in eight different orientations. © By author.

| | | | Annual Electric End-Use | | | Annual Fuel End-Use | |
|---|---|---|---|---|---|---|---|
| | | | HVAC | Lights | Other | HVAC | Other |
| 1 | South | Single Skin | 54.3% | 18.0% | 27.7% | 77.2% | 22.8% |
| | | Green Skin | 39.8% | 23.7% | 36.5% | 86.4% | 13.6% |
| 2 | Southeast | Single Skin | 53.7% | 18.2% | 28.1% | 81.0% | 19.0% |
| | | Green Skin | 35.5% | 25.4% | 39.1% | 87.0% | 13.0% |
| 3 | Southwest | Single Skin | 58.6% | 16.3% | 25.1% | 79.1% | 20.9% |
| | | Green Skin | 42.2% | 22.8% | 35.0% | 86.6% | 13.4% |
| 4 | East | Single Skin | 50.9% | 19.3% | 29.8% | 85.8% | 14.2% |
| | | Green Skin | 34.6% | 25.7% | 39.7% | 88.1% | 11.9% |
| 5 | North | Single Skin | 35.2% | 25.5% | 39.3% | 88.3% | 11.7% |
| | | Green Skin | 33.3% | 26.3% | 40.5% | 89.0% | 11.0% |
| 6 | Northeast | Single Skin | 42.1% | 22.8% | 35.1% | 87.8% | 12.2% |
| | | Green Skin | 35.0% | 25.6% | 39.4% | 88.4% | 11.6% |
| 7 | Northwest | Single Skin | 51.2% | 19.2% | 29.6% | 87.2% | 12.8% |
| | | Green Skin | 36.7% | 24.9% | 38.4% | 88.4% | 11.6% |
| 8 | West | Single Skin | 57.2% | 16.9% | 26.0% | 83.9% | 16.1% |
| | | Green Skin | 40.2% | 23.5% | 36.2% | 87.9% | 12.1% |

Here, it is shown that all orientations represent the different performances of the green facade in energy consumption. The results of the second simulation are divided into eight tests below, and an annual electric end-use and fuel end-use comparison is made between a single skin (main facade that is the curtain wall) and a green skin (as a second layer that is vegetation) for each test.

Test 1: South Orientation

In the south green facade, annual electricity consumption decreased by about 24%. Energy use intensity (EUI) in the southern green facade increased by about 36.5 MJ/m²/year, and the total annual energy cost decreased by approximately 12.5% (Figures 5 and 6).

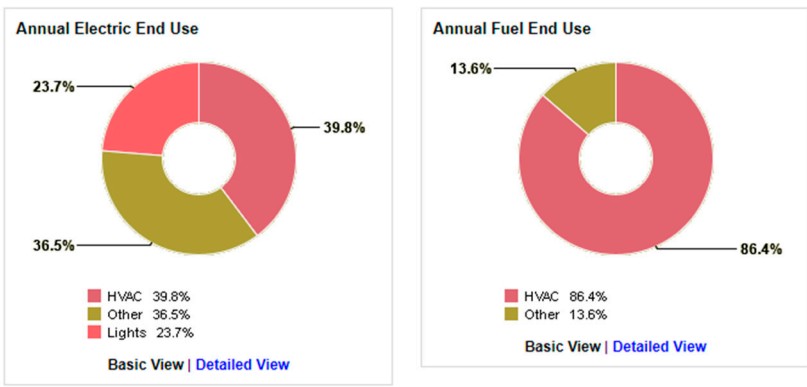

**Figure 5.** Annual electric and fuel end-use for HVAC, lights, and other (miscellaneous equipment) in the southern green facade. © By author.

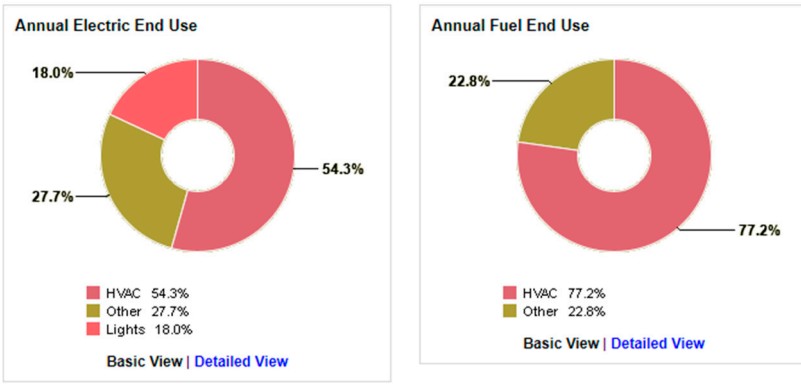

**Figure 6.** Annual electric and fuel end-use for HVAC, lights, and other (miscellaneous equipment) in the southern single facade. © By author.

Test 2: Southeast Orientation

In the southeast green facade, annual electric consumption was reduced by about 28%. The total annual energy cost decreased by approximately 17%, and energy use intensity (EUI) in the southeast green facade decreased by 29.4 MJ/m²/year (Figures 7 and 8).

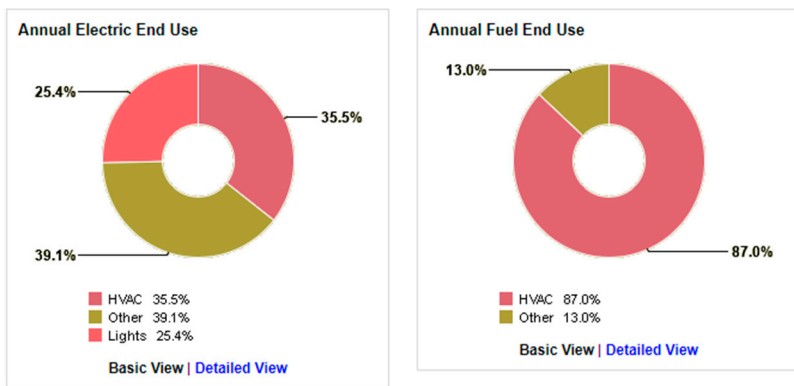

**Figure 7.** Annual electric and fuel end-use for HVAC, lights, and other (miscellaneous equipment) in the southeast green facade. © By author.

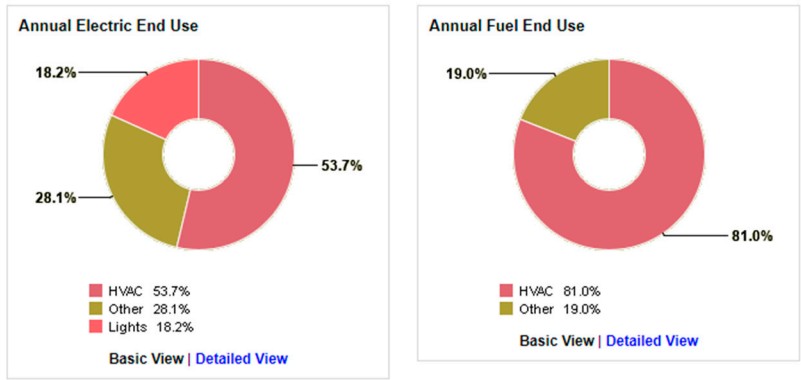

**Figure 8.** Annual electric and fuel end-use for HVAC, lights, and other (miscellaneous equipment) in the southeast single facade. © By author.

Test 3: Southwest Orientation

The southwest green facade showed a 71.2% reduction of annual electrical use. The total annual energy cost was reduced by approximately 17.9%, and energy use intensity (EUI) in this orientation decreased 36.8 MJ/m²/year (Figures 9 and 10).

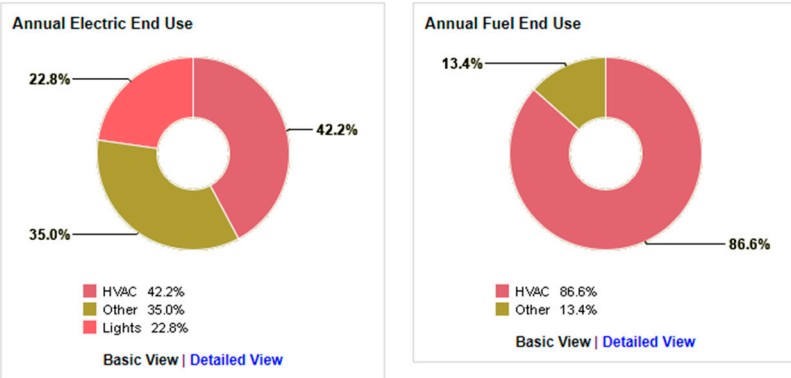

**Figure 9.** Annual electric and fuel end-use for HVAC, lights, and other (miscellaneous equipment) in the southwest green facade. © By author.

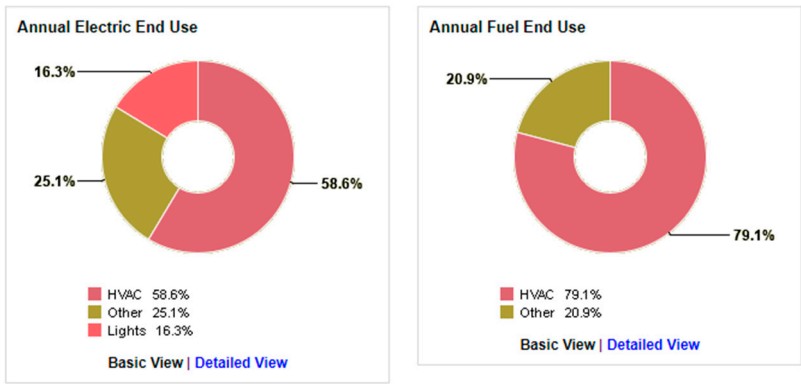

**Figure 10.** Annual electric and fuel end-use for HVAC, lights, and other (miscellaneous equipment) in the southwest single facade. © By author.

Test 4: East Orientation

For the eastern green facade, annual electricity consumption decreased by about 25%. Energy use intensity (EUI) in the east green facade fell by about 71.8 MJ/m²/year, and the total annual energy cost decreased by approximately 16% (Figures 11 and 12).

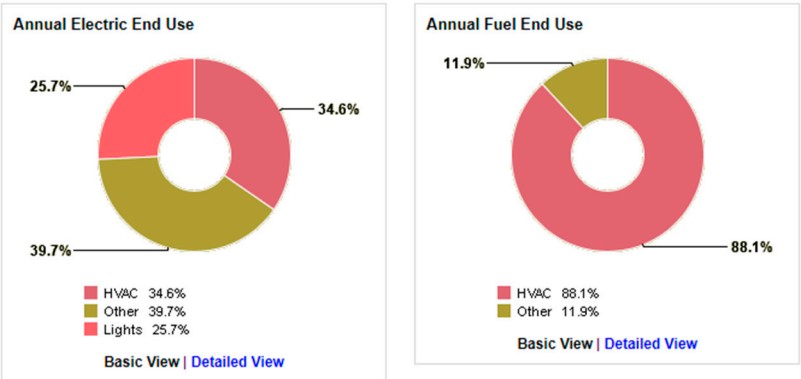

**Figure 11.** Annual electric and fuel end-use for HVAC, lights, and other (miscellaneous equipment) in the eastern green facade. © By author.

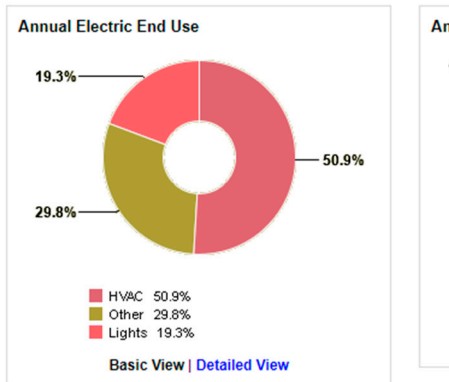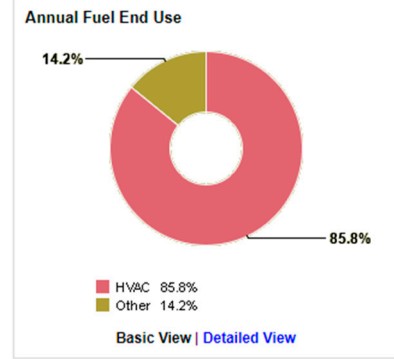

**Figure 12.** Annual electric and fuel end-use for HVAC, lights, and other (miscellaneous equipment) in the eastern single facade. © By author.

Test 5: North Orientation

In the north green facade, annual electricity consumption decreased by about 3%. Energy use intensity (EUI) in the north green facade fell by about 20.1 MJ/m$^2$/year, and the total annual energy cost was reduced by just about 0.5% (Figures 13 and 14).

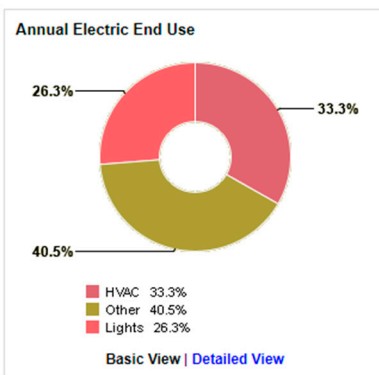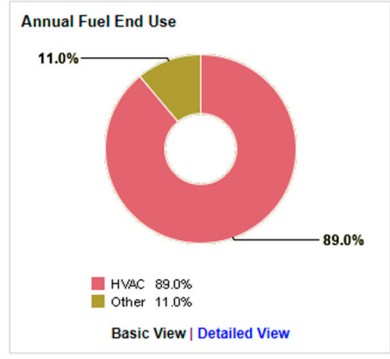

**Figure 13.** Annual electric and fuel end-use for HVAC, lights, and other (miscellaneous equipment) in the northern green facade. © By author.

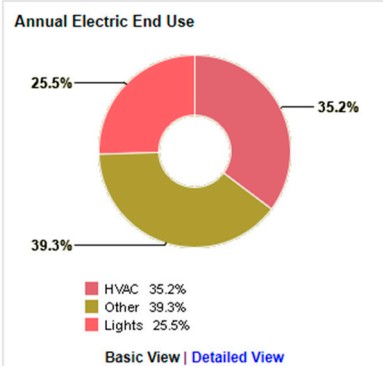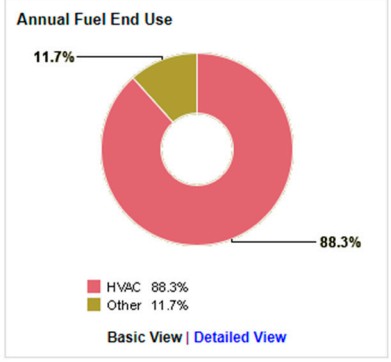

**Figure 14.** Annual electric and fuel end-use for HVAC, lights, and other (miscellaneous equipment) in the northern single facade. © By author.

Test 6: Northeast Orientation

In the northeast green facade, annual electricity use was reduced by about 11%. The total annual energy cost was decreased by just about 7%, and energy use intensity (EUI) in the northeast green facade fell by about 30 MJ/m²/year (Figures 15 and 16).

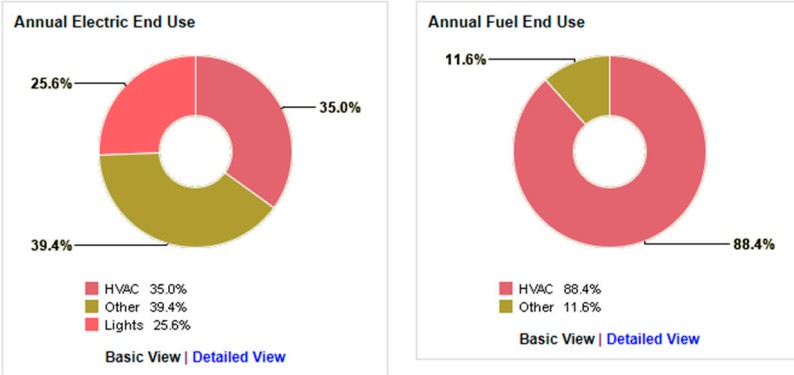

**Figure 15.** Annual electric and fuel end-use for HVAC, lights, and other (miscellaneous equipment) in the northeast green facade. © By author.

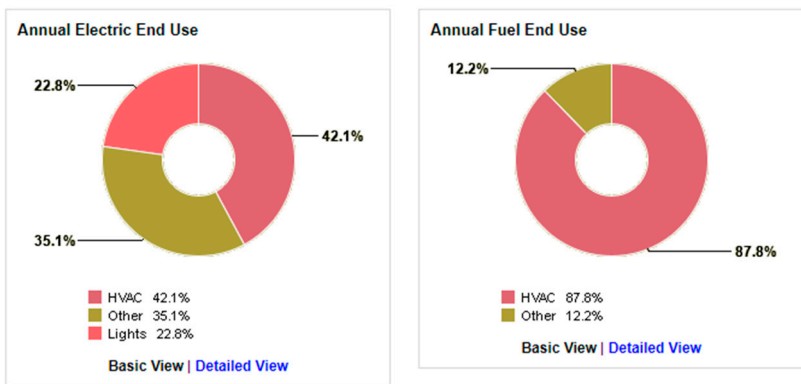

**Figure 16.** Annual electric and fuel end-use for HVAC, lights, and other (miscellaneous equipment) in the northeast single facade. © By author.

Test 7: Northwest Orientation

The northwest green facade showed a reduction in annual electricity consumption, which decreased by about 23%. The total annual energy cost decreased by just about 16%, and energy use intensity (EUI) in the northwest green facade fell by about 97.3 MJ/m²/year (Figures 17 and 18).

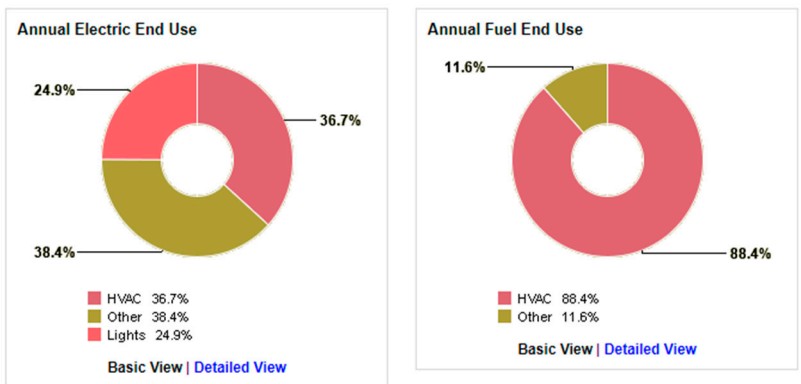

**Figure 17.** Annual electric and fuel end-use for HVAC, lights, and other (miscellaneous equipment) in the northwest green facade. © By author.

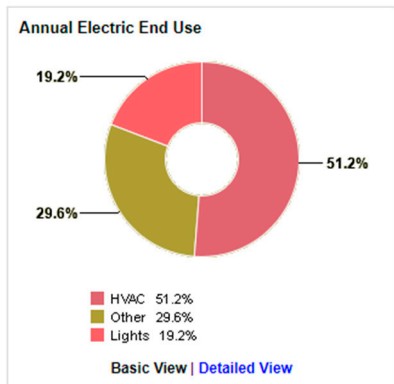
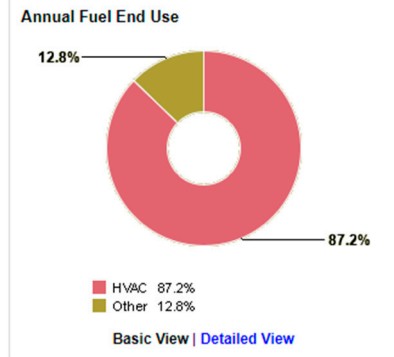

**Figure 18.** Annual electric and fuel end-use for HVAC, lights, and other (miscellaneous equipment) in the northwest single facade. © By author.

Test 8: West Orientation

In the west green facade, annual electricity use decreased by about 28.3%. The total annual energy cost was reduced by about 18.5%, and energy use intensity (EUI) fell by about 73.6 MJ/m²/year (Figures 19 and 20).

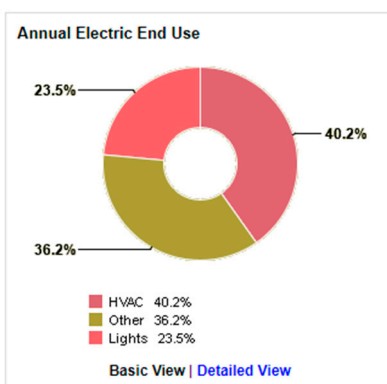
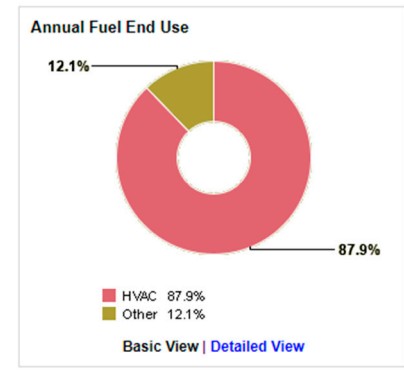

**Figure 19.** Annual electric and fuel end-use for HVAC, lights, and other (miscellaneous equipment) in the west green facade. © By author.

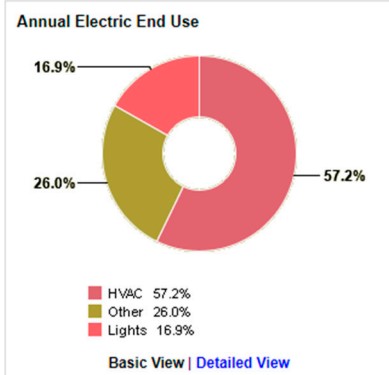
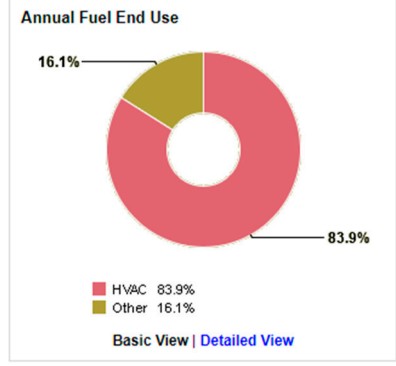

**Figure 20.** Annual electric and fuel end-use for HVAC, lights, and other (miscellaneous equipment) in the west single facade. © By author.

## 4. Conclusions

These results confirm that building orientation, as well as the geographical location and its climate, is a basic requirement for the green facade. It is important to consider the solar radiation quantity that the green facade receives, as it affects the thermal load and controls the thermal behavior and the amount of thermal comfort of the space [31]. In this study, green facades as a second layer were found to change the building behavior in response to solar radiation. This means that in the summer, as well as spring and autumn, occupants could cut down their use of electricity for cooling, therefore allowing the total energy consumption to be reduced significantly. As mentioned in the discussion, according to the simulation of the green facade in different orientations, the northern- and western-orientated green facades' performances were lower than those of facades in other orientations, while the southeast- and southwest-orientated green facades' performances were remarkable as their energy consumption was reduced by about 28%. Furthermore, for the southeast orientation, the total annual energy cost decreased by about 28%; for the southwest orientation, this decrease was 18%. In addition, the selection of an appropriate orientation for the green facade can affect the quantity of ventilation across the inside of the building, which consequentially affects the quantity of energy consumed.

**Author Contributions:** Conceptualization, F.B.M. and I.N.D.; methodology, F.B.M. and I.N.D.; software, F.B.M.; validation, J.M.F.M, I.N.D., and E.R.D.; formal analysis, F.B.M.; investigation, F.B.M.; resources, F.B.M.; data curation, F.B.M.; writing—original draft preparation, F.B.M.; writing—review and editing, F.B.M., J.M.F.M, I.N.D., and A.B.Y.; visualization, F.B.M.; supervision, J.M.F.M and I.N.D.; project administration, F.B.M.; funding acquisition, E.R.D. All authors have read and agreed to the published version of the manuscript.

**Funding:** This research was supported by the National Program of Research, Development and Innovation aimed to the Society Challenges with the references BIA2016-77464-C2-1-R & BIA2016-77464-C2-2-R, both of the National Plan for Scientific Research, Development and Technological Innovation 2013-2016, Government of Spain, titled "Gamificación para la enseñanza del diseño urbano y la integración en ella de la participación ciudadana (ArchGAME4CITY)" and "Diseño Gamificado de visualización 3D con sistemas de realidad virtual para el estudio de la mejora de competencias motivacionales, sociales y espaciales del usuario (EduGAME4CITY)"; (AEI/FEDER, UE).

**Conflicts of Interest**: The authors declare no conflict of interest.

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
