# Peer review of "Building Orientation in Green Facade Performance and Its Positive Effects on Urban Landscape Case Study: An Urban Block in Barcelona"

_sustainability, doi:10.3390/su12219273_

Round 1

Reviewer 1 Report

The title suggests that the paper will analyze multiple benefits (”positive effects”) of green facades in the urban landscape. 

Reading the paper, since the summary, I noticed that it was analyzed only the ”efficiency of green façade in energy consumption” depending on the building orientation.

Even in this case the topic is of interest and can be approached with important results for the implementation of vertical gardens in high-density urban areas, in which terrestrial landscapes cannot be implemented to bring important environmental benefits.

My observations are:

  • References must be improved. There are numerous current studies on this topic. For example, in the introduction there are a number of statements that are not documented - please add current references (last 5-10 years). 
  • In the work you use several terms: vertical gardens, green facades. There is a difference between these terms. Although there are still discussions about these elements of sustainable design, there are extensive studies that clarify the terminology.
  • Please analyze these references:
  • Sustainability 2019, 11, 4579; doi:10.3390/su11174579,
  • Bustami, R.A.; Belusko, M.; Ward, J.; Beecham, S. Vertical Greenery Systems: A Systematic Review of Research Trends. Build. Environ. 2018, 146, 226–237, â
  • Larsen, S.F.; Filippín, C.; Lesino, G. Thermal Simulation of a Double Skin Facade with Plants. Energy Procedia 2014, 75, 1763–1772
  • Maricruz, S.J. Green walls: A Sustainable Approach to Climate Change, a Case Study of London. Archit. Sci. Rev. 2018, 61, 48–57
  • https://www.mdpi.com/2220-9964/9/4/216/htm

Regarding this study, important elements are missing. For example, energy efficiency is influenced not only by the orientation of the building. The following factors can influence this efficiency: the type of plants, the vertical garden system, the degree of maintenance. If the vertical garden has high maintenance requirements, its costs may not justify the implementation - especially if these costs exceed the savings from energy consumption. The orientation of the facades also determines the selection of plant species according to their requirements for environmental conditions and especially for light. The plan species have various requirements for the substrate, vertical garden system, fertigation and maintenance - elements that lead to additional costs. 

In this study were considered green facades with different cavities in terms of thickness. But it is not specified what type of substrate is used, what type of system (because there are very varied variants). These characteristics influence energy efficiency, perhaps even more than the orientation of the facade.

I believe that this study can be a starting point for a deeper research of the subject that takes into account important aspects.

I recommend that you study this topic carefully, develop, and improve the paper to be published.

Author Response

Dear Reviewer,

Please see the revised version.

The article has been modified and respectively presented in below:

  1. Removed abbreviations from the abstract paragraph.
    2. The first paragraph (the aim of the article) moved in the last paragraph in the introduction. (lines 35-39 first manuscript and lines 87-93 of the modification manuscript)
    3. The paragraph about Autodesk Green Building Studio summarized and joined to aim paragraph (lines 40-44 first manuscript)
    4. Added a new paragraph in the introduction part (first paragraph) lines 34-42 of the modification manuscript.
    5. Inserted references at line 46 of the first manuscript (line 44 modification manuscript)
    6. Inserted References at lines 46 and 52 of modification manuscript
    7. Added a new paragraph in the introduction at lines 59-62 modification manuscript
    8. Inserted references at lines 64 and 68 of the modification manuscript
    9. Added a new paragraph in lines 81-86 of the modification manuscript
    10. Modified methodology paragraph, lines 86-88 first manuscript and lines 101-104 of the modification manuscript
    11. Added new data in the case study and scenario descriptions lines 113-116 of the modification manuscript
    12. Changed the size of figure1.
    13. Modified lines 109-113 first manuscript and lines 128-133 of modification manuscript
    14. Changed the size of table 3.
    15. Modified lines 140-143 first manuscript and lines 161-164 of the modification manuscript
    16. Modified lines 147-151 first manuscript and lines 168-173 of the modification manuscript
    17. Modified discussion, lines 154-168 first manuscript and lines 176-191 of the modification manuscript
    18. Changed the size of figures 5 -20.
    19. Modified as much as possible conclusion paragraph.
    20. Completed author contributions paragraph.
    21. Modified references format.

Kindest regards,

Faezeh

Reviewer 2 Report

This paper presented the affecting of building orientation in efficiency of green façade in energy consumption. The study is applicate to an urban block in Passeig de Gracia, in the barrio de L’Eixample in Barcelona. The topic is really interesting, as the green facades are not so much treated in the literature. Thus, it is surely interesting to be published in this special issue of Sustainability. The aim of this research is to investigate the impact of building orientation in green facade on energy consumption that provides the maximum reduction of energy use. The introduction is not an introduction. It expresses the aim of the paper, it describes the software autodesk  and it introduces the important topics connected with this theme. I suggest to insert a section on aims in the methodology section and to insert here the part on the aims. Also, to cut the description of the software. The introduction has to be totally revised, focusing better on the topic of building and urban strategies for the reduction of energy consumptions, considering the building envelope only. In this case, you can compare the performances of infrared coatings (I.e. refer to  https://doi.org/10.1016/j.buildenv.2018.02.034), pcm (I.e refer to https://doi.org/10.1016/j.egypro.2014.12.362) and green facades in historical towns. Then, you can focus on orientation of the green facade, as a main aspect. The methodology is not clear. Here some aspects on aims already inserted in the introduction are present. I suggest to revise it, using bullet points on the different phases. Results and discussion at not enough. Particularly, the discussion is made by a series of graphs with simple explications. They could be inserted in a table, comparing the differences in results. studies data collection and validation. Also, when dealing with energy simulation, the aspects on input data and validation are necessary. I suggest to refer to doi.org/10.1016/j.rser.2019.109509 on input data and to https://doi.org/10.1016/j.buildenv.2020.107081 on validation scheme for energy simulation There are several tips and problem with the txts. The most important aspect is related to the Results, that are not well documented and discussed. Please implement this part. Conclusion not enough, please add here the novelty of your results and its utility for the scientific community. Also, the discussion using different points permits to focus on the most important results of you research. Reference are old, not scientific in many case. Please, implement it with new references by the literature review. I try to suggest few papers to enlarge the discussion.

Author Response

(The authors gave the same response as above.)

Reviewer 3 Report

In the submitted manuscript the affecting of building orientation in efficiency of green façade in energy consumption is investigated as a case study. The manuscript is in general well organized and it would be of interest to the research community in this field of work as it offers results that are of not only theoretical value, but also useful in practice. For instance, the review and the drawn conclusions would support developers and thermal engineers in their work.

In order to improve the readability and clarity of the manuscript, a few minor remarks need to be addressed before the paper is to be accepted for publishing:

1) The abbreviations (as CO2, EUI…etc.) in the abstract section should be avoided!

2) The quality of Figure 1 and Table 3 should be increased or edited again.

3) The font size on Figures 5-20 should be increased for better readability!

4) The introduction section could be extended more. There are many other research results in this field that should be mentioned by the authors as research backgrounds in the “Introduction” section of the paper. By this way the “Introduction” and also the “References” sections of this paper should be completed with the under mention relevant references especially that relates to this field:

[3] Furundžić, A.K., Vujošević, M., Petrovski, A.. Energy and environmental performance of the office building facade scenarios. Energy 2019, 183, 437-447.

[4] Pérez, G., Coma, J., Sol, S., Cabeza, L.F. Green facade for energy savings in buildings: The influence of leaf area index and facade orientation on the shadow effect. Applied Energy 2017, 187, 424-437.

Please put these references into the text of the following added sentence (in line 62):

A building with the right orientation can multiply the efficiency of the green façade as a second layer in façade [3-4].

Please complete the References section of your paper with these referred papers: with numbers [3]; [4] references!

5) The reference style of the “References“ section of the recent paper does not meet with the requirements of the journal. Please check the relating formal requirements in the “guide for authors” again and correct it following the instructions!

To improve the paper based on these minor modifications are very significant to have success in acceptance for publication!

Thank you for your consideration in advance!

Author Response

(The authors gave the same response as above.)

Round 2

Reviewer 1 Report

Dear authors,

the structure of the revised paper is much improved and allows easy understanding of the research and its purpose.

The bibliographic references support the scientific approach of the research, the introduction is relevant.

However, I have noticed some inconveniences that require your intervention:

  • First of all, a revision of some sentences is needed - from the perspective of the English language. I understand what you want to express because I am familiar with the subject of the paper and I know extensively the aspects involved in green facades. But I had to read some sentences twice and analyze the content very carefully. 
  • Please check the following lines:
  • Row 38-39 - ”and being dramatically applied” - is not a scientific expression, in what way it is dramatically applied? you meant that: it is applied extensively - on a large scale?
  • Row 60-62 - a confusing and incorrect phrase - please rephrase;
  • Row 74-75 - please rephrase and correct;
  • Row 78 - insert ”are” between ”requirements” and ”summarized”;
  • Row 105-108 - It requires a reformulation and a detailed explanation of the method. I assume that when you say "runs" you are referring to the variants / variables introduced in the study/in simulation. You should describe this more carefully - I think it is important for the method you use;
  • Row 110-116 
  • Row 120-121
  • Row 254-255
  • I have tried to list all the lines that require the correction of the English language style  - but I have come to the conclusion that you must correct almost the whole paper.
  • In conclusion, for this variant, the corrections refer especially to the expression in English of your ideas.

Author Response

Dear reviewer,

Thank you for your comments. I modified the article according to your report.
The changes are as follows:

  1. First of all, the article has edited extensively by the English native speaker.
  2. Row 38-39, revised sentence. (in the modified article, row 38-40)
  3. Row 60-62, rephrased sentence and modified it. (in the modified article, row 65-68)
  4. Row 74-75, rephrased, and corrected. (in the modified article, row 81-82)
  5. Row 78, inserted “are” between ”requirements” and ”summarized”. (in the modified article, row 85-86)
  6. Row 105-108, has been modified throughout the paragraph. (in the modified article, row 113-117)
  7. Row 110-116, has been modified. (in the modified article, row 119-124)
  8. Row 120-121, has been modified. (in the modified article, row 128-129)
  9. Row 254-255, has been modified. (in the modified article, row 264-265)

Respectfully,

Faezeh Bagheri.M

Reviewer 2 Report

the paper was improved comparing to the previous version, but the response of author to my comments is very general. I can’t understand in which way you reply to my comments. The reply is very punctual but  in the most part the modifications are not corresponding to my comments. Please, revise it again and reply comment by comment (also reporting the comment you want to reply). In this way I can evaluate the modifications. 

Author Response

Dear reviewer,

I write to express respect for your comments in the review round 1. I modified according to your reports exactly, but I didn't know I have to explain in detail for reviewers' reports individually.

The changes of the round 1 are as follows:

1- The introduction part has been modified and has been totally revised.
2- The paragraph about Autodesk Green Building Studio summarized and joined to aim paragraph (lines 40-44 first manuscript)
3- The infrared coatings are not related to this article and in this article focused on the aspect of building orientation on the green facade performance.
4- The methodology part has been modified.
5- In the result section, there are 3 tables that showed the comparison of the simulation data, and also in the discussion section as well. Because of the size of the figures, I was not able to collect them in one table. As a result, I made the sections for each simulation to explain the details of them completely.
6- The methodology section has been modified.
7- About input data and validation scheme that you recommended using (doi.org/10.1016/j.rser.2019.109509 and https://doi.org/10.1016/j.buildenv.2020.107081), that were actually useful for my future article which is about the thermal aspect of the green facade on the building. However, that were not related to this article because it was about hygrothermal simulations and the behavior of historical buildings.
8- The conclusion section has been modified.
9- References have been updated.

I appreciate you for considering my article.

Respectfully,

Faezeh Bagheri.M

Round 3

Reviewer 1 Report

Dear authors,

the paper is improved. I agree with the changes you have made, and I consider that the paper can be published in this form.

Author Response

Dear Reviewer,

Thank you for your report. I want to let you know about the final modifying according to the second reviewer's report.
In the discussion section, added a table of annual electric and fuel end-use comparison (line198).

Kindest regards,
Faezeh

Reviewer 2 Report

The manuscript has improved in the introduction. It is not easy to read, especially in the part related to the discussion. Here there are several graphs that can be summarized in a table to compare the results in a easy way. Aldi, it is mandatory to comment ant to compare the results obtained. 

Author Response

Dear Reviewer,

Thank you for your report. In the discussion section, added a table of annual electric and fuel end-use comparison (line198).

Kindest regards,
Faezeh
